

# Humans differ in their personal microbial cloud

James F. Meadow[1,2], Adam E. Altrichter[1,2], Ashley C. Bateman[1,2], Jason Stenson[1,3], GZ Brown[1,3], Jessica L. Green[1,2,4] and Brendan J.M. Bohannan[1,2]

[1] Biology and the Built Environment Center, University of Oregon, Eugene, OR, USA
[2] Department of Biology, Institute of Ecology and Evolution, University of Oregon, Eugene, OR, USA
[3] Department of Architecture, Energy Studies in Buildings Laboratory, University of Oregon, Eugene, OR, USA
[4] Santa Fe Institute, Santa Fe, NM, USA

Corresponding author
James F. Meadow,
jfmeadow@gmail.com

## ABSTRACT

Dispersal of microbes between humans and the built environment can occur through direct contact with surfaces or through airborne release; the latter mechanism remains poorly understood. Humans emit upwards of $10^6$ biological particles per hour, and have long been known to transmit pathogens to other individuals and to indoor surfaces. However it has not previously been demonstrated that humans emit a detectible microbial cloud into surrounding indoor air, nor whether such clouds are sufficiently differentiated to allow the identification of individual occupants. We used high-throughput sequencing of 16S rRNA genes to characterize the airborne bacterial contribution of a single person sitting in a sanitized custom experimental climate chamber. We compared that to air sampled in an adjacent, identical, unoccupied chamber, as well as to supply and exhaust air sources. Additionally, we assessed microbial communities in settled particles surrounding each occupant, to investigate the potential long-term fate of airborne microbial emissions. Most occupants could be clearly detected by their airborne bacterial emissions, as well as their contribution to settled particles, within 1.5–4 h. Bacterial clouds from the occupants were statistically distinct, allowing the identification of some individual occupants. Our results confirm that an occupied space is microbially distinct from an unoccupied one, and demonstrate for the first time that individuals release their own personalized microbial cloud.

## INTRODUCTION

Humans harbor diverse microbial assemblages in and on our bodies (*HMP Consortium, 2012*), and these distinctly human-associated bacteria can be readily detected inside of buildings on surfaces, in dust, and as bioaerosols (*Hospodsky et al., 2012*; *Täubel et al., 2009*; *Fierer et al., 2010*; *Flores et al., 2011*; *Flores et al., 2013*; *Meadow et al., 2013*; *Kembel et al., 2012*; *Kembel et al., 2014*). Human-associated bacteria disperse into and throughout the built environment by three primary mechanisms: (1) direct human contact with indoor surfaces; (2) bioaerosol particle emission from our breath, clothes, skin and hair; and (3)

resuspension of indoor dust containing previously shed human skin cells, hair and other bacteria-laden particles. Recent studies of built environment microbes have largely focused on direct human contact with surfaces, and demonstrated that people can leave behind bacterial signatures indicative of particular body parts and types of bodily contact (*Flores et al., 2011*; *Flores et al., 2013*; *Meadow et al., 2014*). Given the long-term identifiability of the human microbiome (*Franzosa et al., 2015*), in some cases these bacterial assemblages can even be traced back to an individual person (*Fierer et al., 2010*), though this has only been demonstrated using direct contact.

Bioaerosols (airborne biological particles including bacteria and bacteria-laden particles) have often been the focus of infection-control studies, but their role in seeding the built environment microbiome is poorly understood. Research on bioaerosols to date has rarely focused on the direct emission of bioaerosols from human sources, because it is difficult to disentangle direct emission from resuspension of dust observationally (*Meadow et al., 2013*; *Qian et al., 2012*; *Qian, Peccia & Ferro, 2014*; *Nazaroff, 2014*; *Hospodsky et al., 2014*; *Bhangar et al., 2015*; *Adams et al., 2015*). Previous attempts to account for dust resuspension in indoor bioaerosols indicate that active human emission, as opposed to resuspended particulate matter, is an underestimated part of the total indoor airborne bacterial pool in buildings (*Qian, Peccia & Ferro, 2014*). For instance, several recent studies (*Hospodsky et al., 2012*; *Hospodsky et al., 2014*; *Bhangar et al., 2015*; *Adams et al., 2015*) detected substantial particle concentrations from a group of occupants in a room, as well as a single occupant in a room, even after controlling for resuspended dust. Since humans shed approximately $10^6$ particles ($>0.5 \, \mu m$ diameter) per hour (*You et al., 2013*; *Bhangar et al., 2015*), and many of these likely contain bacteria, the actively shed human microbial contribution is thought to play a substantial role in seeding the built environment microbiome. Additionally, microbes from occupants in a new home can be detected inside the house in a matter of days (*Lax et al., 2014*), illustrating the magnitude of airborne flux from humans to their built environment. This flux potentially also mediates interactions with other humans and their associated microbiota within dispersal range.

Human interactions with indoor airborne microbes have been investigated for more than a century (*Tyndall, 1881*; *Carnelley, Haldane & Anderson, 1887*; *Tyndall, 1876*), although almost exclusively from the perspective of disease and airborne-transmission of pathogens (*Noble et al., 1976*; *Sherertz, Bassetti & Bassetti-Wyss, 2001*; *Tang et al., 2011*). Human interactions with non-pathogenic microbes have recently received increased attention for their integral roles in healthy human function (*HMP Consortium, 2012*). We are just beginning to understand how these interactions structure the human microbiome, including interactions with indoor bioaerosols and indoor dust (*Fujimura et al., 2010*; *Fujimura et al., 2013*). It is plausible that direct emission of bacterial cells from an individual results in a detectable human microbiome signal that is traceable to a particular individual, similar to what has been reported after contact with indoor surfaces (*Fierer et al., 2010*), although this has never been demonstrated. Recently emitted bioaerosols potentially represent a distinct microbial source pool if they contain bacterial taxa that are unable to persist in desiccated dust or on surfaces, and therefore might not be otherwise detected.

These recently emitted microbes might also more readily colonize other humans within the built environment than those found on surfaces or in resuspended dust, since recently emitted microbes are more likely to be physiologically active and have not been subjected to prolonged desiccation or UV exposure before colonization can successfully occur.

In order to understand the human contribution to bioaerosols within built environments, and the extent to which this emitted bioaerosol pool contributes to the residual human-microbial signal detected in indoor dust and on surrounding indoor surfaces, we characterized the airborne bacterial cloud of a person sitting in a sanitized experimental climate chamber (Fig. S1). Background bacterial biomass in the chamber was reduced by a combination of surface disinfection and ventilation control. Over the course of two separate experiments, we used high-throughput DNA sequencing methods to characterize airborne bacterial community composition emitted by 11 different human occupants. During the first experiment, we compared airborne bacterial assemblages to those detected simultaneously in an identical, adjacent, unoccupied side of the chamber. This was repeated for three different people, each for 4- and 2-hour sampling periods. To assess the potential for these airborne particles to result in a detectable human microbiome signal as settled dust on surrounding surfaces, we sequenced DNA from settling dishes in each sampling period, and compared those to airborne assemblages. Given that occupants could each be clearly detected and differentiated from one another, we designed a second experiment to further explore the distinguishability of the personal microbial cloud. For this we sampled 8 different people for 90 min each, and with air flowing at 1 air change per hour (ACH) and 3 ACH. Each occupant's personal microbial emissions were compared among occupants, and to filtered supply and exhaust air from the occupied chamber to assess personal detectability within a building's ventilation system.

## METHODS AND MATERIALS

### Experimental design

All samples were collected over December 5–7, 2012 (first experiment), and April 22–May 3, 2013 (second experiment), at the Energy Studies in Buildings Laboratory, Portland, OR, USA, using the custom Climate Chamber (Fig. S1; interior dimensions = c. 3.7 m long × 2.4 m wide × 2.9 m high; 25.75 m$^3$). Filtered air was supplied through a ceiling plenum, and exhausted through a floor plenum. During the first experiment, the chamber floor and walls were lined with 0.15 mm clean-room plastic sheeting (Visqueen, British Polythene Ltd., Heanor, UK). The sheeting was anti-static and fire retardant, and was washed and double bagged by manufacturers. Sheeting edges were sealed to walls, floors and ceilings with vinyl cleanroom tape (UltraTape, Wilsonville, Oregon, USA). A vertical partition of plastic sheeting was taped down the middle of the chamber to create two identical chambers for simultaneous sampling. This simultaneous design allowed us to monitor any changes to the chamber air not due to the occupant over the course of the first experiment. During the second experiment, the plastic was not used, nor was the chamber divided, and thus the total volume of air in the occupied chamber was doubled for the second experiment.

Continuous fan powered supply air was used to maintain positive pressure in the chamber throughout both experiments; this helped to reduce the introduction of background bioaerosols from outside the chamber. Conditioned indoor air (first experiment) or outdoor air (second experiment) passed through a MERV-15 (first experiment) or MERV-8 (second experiment) filter to reduce dust and bacterial introduction into the chamber. However the ductwork was not sterilized after the point of filtration, nor was the inline mechanical equipment. Thus, the chamber was not fully sealed to unfiltered air. Ducted supply air was diffused through a ceiling plenum into the chamber. Air was exhausted (second experiment) through the floor plenum under the chamber, and through an exhaust duct. During the second experiment, vacuum air filter samplers were installed in-line to collect supply air after the supply fan and MERV-8 filter, and also in the exhaust duct.

Air exchange rates were monitored throughout both experiments by balancing supply and exhaust air velocities, measured in center-of-duct point with a multi-function ventilation meter and a thermoanemometer probe (#9555 and #964, respectively; TSI Incorporated, Shoreview, MN). Air pressure within the chamber was measured with a differential air pressure transducer (#T-VER-PXU-L; Onset Computer Corporation, Bourne, Massachusetts, USA).

All interior chamber surfaces, including plastic lining and sampling apparatuses, were washed with bactericidal chemical treatment (Cavicide, Metrex Research, Orange, California, USA) before and between each experimental trial to reduce the microbial load in the chamber and to limit cross-contamination between occupant treatments. The occupant chairs used throughout both experiments were Caper Stacking Chairs (Flexnet set, fixed arms, hard-floor casters; Herman Miller, Zeeland, Michigan, USA). All technical staff entering the chamber for cleaning or sample collection wore sterile, hooded Tyvek garments, gloves and face-masks.

All eleven subjects were free of disease symptoms at the time of sampling, had not taken antibiotics for at least 4 months, and were between the ages of 20 and 33. The subjects were informed as to the full nature and design of the study and gave written consent to be participants. All research protocols were approved by the University of Oregon Institutional Review Board (protocol # 03172014.021). Identities of participants were never recorded on samples or in resulting datasets.

Standard bioaerosol and settled particle sampling protocols used by *Bowers et al. (2011)*, *Bowers et al. (2012)*, and *Adams et al. (2013)* were modified for our experimental climate chamber approach. Each occupant sat in a disinfected plastic rolling chair surrounded by 12 sterile 0.2 μm cellulose-nitrate air filters that were arranged equidistant from the seat (Fig. S1). Air filters were arranged in rings of 6 samplers each: one ring at shoulder height when seated (1 m) and the other just above floor height (15 cm). During the second experiment, one ring of 6 filters were placed at shoulder height, but not at floor height. Air was drawn through the filters by a pump external to the chamber at c. 10 L min$^{-1}$ for the first experiment, and c. 24.5 L min$^{-1}$ for the second. Settled particles were collected on both the lid and base of 6 empty, sterile petri dishes (15 cm) that were placed face-up on the floor in a circular pattern surrounding the occupant. In order to expose each air filter

to the occupant in a similar manner, the occupant was instructed to rotate approximately 60° on regular intervals to face a new filter or pair of filters; occupants sat in rolling chairs, so rotating required minimal movement within the chamber. The unoccupied chamber was arranged in identical fashion, with the exception that the occupant in the occupied chamber was allowed a ethanol surface-sterilized laptop for entertainment and as a means of communicating with those outside of the chamber; in order to reduce heat generated by the laptop computer, occupants held the laptop on a rubberized, surface-sterilized lap-desk. The laptop and lap-desk were absent from the unoccupied chamber.

Occupants self-reported their comfort, and any necessary temperature adjustments were made without tempering air, but rather by adjusting radiant floor temperature. Air temperature and relative humidity were monitored using data loggers (#U12-012; Onset Computer Corporation, Bourne, Massachusetts, USA). Temperatures inside the climate chamber throughout both experiments ranged from 22 to 26 °C, and relative humidity ranged from 25 to 45%.

During the first experiment, each day consisted of a single occupant in the chamber for one 240- and one 120-min sampling period, with a break between sampling periods. During the second experiment, each occupant was in the chamber for two separate 90-min periods, once at 3 air changes per hour (ACH) ventilation rate, and again for 1 ACH.

Particle count data was collected at a rate of 2.83 L min$^{-1}$ in 1 min intervals, and size fractionated with the AeroTrack 9306-V2 (TSI Inc., Shoreview, Minnesota, USA). Three different particle size classes (2.5–5 μm, 5–10 μm and 10+ μm) were considered for this study. All particle counts were averaged over 10 min intervals (5 for the second experiment) and converted to L min$^{-1}$ ratio above simultaneous unoccupied values for the first experiment. Since no unoccupied chamber was used in the second experiment, we calculated a particle deposition loss coefficient (*Tracy et al., 2002*) by comparing particle counts in the occupied chamber to particle counts in the supply duct system. Filters and settling dishes were immediately packaged, transported on dry ice, and stored at −80 °C until further processing.

## 16S library preparation and sequencing

To avoid confounding effects introduced during library preparation, all samples were randomized across extraction batch, amplification batch, and processing order. Air filters and settling dishes from both experiments were all processed using methods specifically for low-biomass samples adapted from *Kwan et al. (2011)*, and amplicon libraries were constructed following methods from *Caporaso et al. (2012)*, and *Fadrosh et al. (2014)*.

### *First experiment*

Particles and nucleic acids from the lid and base of each settling dish were collected and combined using a PBS-moistened, DNA-free cotton swab (Cat. 25-806 1WC FDNA; Puritan Medical, Guilford, Maine, USA) to wipe the lid and base twice-over on each, in perpendicular directions and rotating the swab a quarter turn with each pass. Cells and nucleic acids were then eluted from the air filters and swabs by vortexing each sample in 4 mL of sterile PBS (biotechnology grade; Amresco, Solon, Ohio, USA). The eluate was

subsequently concentrated to approximately 500 μL in an Amicon Ultra-4 centrifugal filter (30 kDa). 200 μL of this concentrated sample was extracted using the MoBio htp-PowerSoil DNA Isolation kit according to the manufacturer's specifications with the following modifications: samples were individually bead beat in 2 mL collection tubes with 0.1 mm glass beads, 200 μL of phenol:chloroform:IAA was added to the bead tube prior to beat beating, tubes were beat using a FastPrep1200 homogenizer at setting 5.0 for 40 s, solutions C2 and C3 were added together in equal volumes, solution C4 and absolute ethanol were added in equal volumes to the lysate, 650 μL of absolute ethanol was used to wash the spin column prior to solution C5, and DNA was eluted in 70 μL of elution buffer.

Sequencing libraries were prepped using a modification of the *Caporaso et al. (2012)* protocol wherein 16S rRNA gene primers 515F and Golay-barcoded 806R were used in triplicate PCRs per sample, followed by equivolume combination of all samples, and concentrated to 25 μL (Zymo Research Clean and Concentrate-5). This was followed by gel electrophoresis size selection and extraction of the pooled samples (Qiagen MinElute Gel Extraction), with a final clean up step (Zymo Research Clean and Concentrate-5). The PCR had the following components (25 μL total volume): 13.25 μL DNA-grade water, 5 μL 5× HF Buffer, 0.5 μL dNTPs (10 mM), 0.5 μL each primer (10 μM), 0.25 μL Phusion Hot Start II polymerase (2 U/μL), and 5 μL of genomic DNA template. The PCR was carried out under the following conditions: an initial denaturation step of 98 °C for 1 min, followed by 35 cycles of 98 °C for 20 s, 52 °C for 30 s, and 72 °C for 30 s, with a final extension at 72 °C for 10 min. The final library was then sent to the Dana-Farber Cancer Institute Molecular Biology Core Facilities for 250 PE sequencing on the Illumina MiSeq platform.

### Second experiment

Air samples from the second experiment were processed using the following modifications to the MoBio PowerSoil-htp kit: filters were bead beat for 1.5 min at maximum speed with the FastPrep1200 homogenizer and heated for 10 min at 65 °C, Solutions C2 and C3 were omitted prior to loading on the spin-column, and samples were eluted into 70 μL.

Each sample was amplified with a 2 step PCR prep method for Illumina sequencing of the V3-V4 region with 319F and 806R dual indexed primers including heterogeneity spacers to improve low plexity libraries, similar to methods used by Fadrosh and colleagues (*2014*). PCR1 was run in triplicate for each sample and included 11.75 μL PCR-grade water, 0.25 μL Phusion HS II polymerase (2 U/μL), 5 μL 5× HF buffer, 0.5 μL dNTPs, 2.5 μL forward and reverse gene primer mix (5 μM each) with heterogeneity spacers, and 5 μL template genomic DNA. PCR1 was run under the following conditions: 98 °C for 2 min, followed by 25 cycles of 98 °C for 20 s, 50 °C for 30 s, and 72 °C for 30 s, with a final extension of 2 min at 72 °C. Triplicates were pooled and cleaned with Qiagen MinElute 96 UF PCR Purification kit prior to PCR2. PCR2 contained 6.75 μL PCR-grade water, 0.25 μL Phusion HS II polymerase (2 U/μL), 5 μL 5x HF buffer, 0.5 μL dNTPs, 1.25 μL of forward and reverse primers (10 μM each) with Illumina adapter and index sequences, and 10 μL template from cleaned PCR1 products. PCR2 was run under the following conditions: 98 °C for 1 min, followed by 10 cycles of 98 °C for 20 s, 63 °C for 30 s, and 72 °C for 30 s, with a final extension of 3 min at 72 °C. Samples were then cleaned,

multiplexed and concentrated to be run on a 1% gel for size-selection, then underwent a final clean-up step with Zymo Research Clean and Concenrate-25 kit. The final library was submitted for 300 PE sequencing run on the Illumina MiSeq platform at the Molecular Biology Core Facility at Harvard's Dana Farber Cancer Institute.

## Data processing and statistical analysis

Raw sequences from the first experiment were processed using a QIIME v. 1.7 pipeline (*Caporaso et al., 2010*). We retained and demultiplexed $1.007 \times 10^7$ sequences with an average quality score of 30 over 97% of the sequence length. Sequences were binned into OTUs at 97% similarity using UCLUST denovo clustering (*Edgar, 2010*), which resulted in c. $2.4 \times 10^5$ OTUs across 300 samples. Raw sequences from the second experiment were processed using the QIIME v. 1.8 pipeline, except that OTUs were clustered using USEARCH v. 7 (*Edgar, 2013*). We retained and demultiplexed $7.5 \times 10^7$ sequences with expected error rates less than 0.5. Taxonomy was assigned to OTUs using the RDP classifier and Greengenes version '4feb2011' core set (*DeSantis et al., 2006*).

After quality filtering, demultiplexing, and OTU clustering, all statistical analyses were conducted in R (*R Development Core Team, 2010*), primarily with the vegan, labdsv and ape packages (*Oksanen et al., 2011*; *Roberts, 2010*; *Paradis, Claude & Strimmer, 2004*). Plant chloroplast and mitochondrial sequences were removed from both datasets prior to analysis. Apparent contaminants were also analyzed separately for their influence on results, and those exerting influence were removed from downstream analysis (4 OTUs from the first experiment, and 10 from the second).

### *First experiment*

All samples in the first experiment were rarefied to 1,000 sequences per sample to achieve approximately equal sampling depth. $\beta$-diversity was calculated using the Canberra taxonomic metric, and ordinations were constructed using iterative non-metric multi-dimensional scaling (NMDS). Community differences were assessed using permutational multivariate analysis of variance tests (PERMANOVA). Since community differences were tested with permutational tests, we report $p$-values down to, but not below, 0.001. Clustering was conducted with an average linkage method based on Canberra distances. Indicator species analysis followed *Dufrêne & Legendre (1997)*, and OTUs were further investigated if uncorrected $p$-values were less than 0.05. The most significant indicator OTUs from 4-hour air filters (indicator value >0.6 and $p$-value <0.001) are shown in Table S1. Representative sequences from each OTU were BLAST'ed against the NCBI 16S isolate database, resulting in putative species assignments and NCBI accession numbers.

### *Second experiment*

The goal of the second analysis was different from the first. We were primarily interested in the subset of OTUs that help to distinguish each occupant, and not in the OTUs that were abundant in both the unoccupied and occupied air. These targeted OTUs were selected based on their GreenGenes taxonomic assignments: *Corynebacteriaceae, Staphylococcaceae, Streptococcaceae, Lactobacillaceae, Propionibacteriaceae, Peptostreptococcaceae, Bifidobacteriaceae, Micrococcaceae, Carnobacteriaceae, Dietziaceae, Aerococcaceae*, and *Tissierellaceae*.

**Table 1 During the first experiment, occupied chambers were significantly different from unoccupied across every 4-hour sampling period, regardless of sampling method.** The tests for *"occ vs. unocc"* consider two different groups, combining all occupied samples and all unoccupied samples, whereas *"3 people vs. unocc"* considers 4 different groups including 3 separate occupants and all unoccupied samples together.

| Data subset | Test | $n$ | $R^2$ | $p$[a] |
|---|---|---|---|---|
| All samples | Occ vs. unocc | 211 | 0.01 | 0.001 |
| All 4-hour | Occ vs. unocc | 106 | 0.017 | 0.001 |
| All 2-hour | Occ vs. unocc | 105 | 0.014 | 0.001 |
| All air filters | Occ vs. unocc | 140 | 0.014 | 0.001 |
| All settling dishes | Occ vs. unocc | 71 | 0.018 | 0.001 |
| **Only 4-hour samples** | | | | |
| Air filters | Occ vs. unocc | 71 | 0.023 | 0.001 |
| Air filters | 3 people vs. unocc | 71 | 0.061 | 0.001 |
| Air filters | Occupants | 36 | 0.078 | 0.001 |
| Air filters | Only unoccupied | 35 | 0.061 | 0.027 |
| Settling dishes | Occ vs. unocc | 35 | 0.035 | 0.001 |
| Settling dishes | 3 people vs. unocc | 35 | 0.098 | 0.001 |
| Settling dishes | Occupants | 18 | 0.13 | 0.001 |
| Settling dishes | Only unoccupied | 17 | 0.126 | 0.221 |

**Notes.**

[a] Results are from PERMANOVA on Canberra distances.

Thus we didn't rarefy the second dataset, but rather created a subset of relative abundances for analysis. Additionally, since several of these human-associated groups were the most abundant and distinguishing among occupants, we used the Bray-Curtis dissimilarity metric for multivariate analyses, and the Jaccard distance (as a percent of shared OTUs) to show the average shared relationships in Fig. S4. Since community differences were tested with permutational tests, we report $p$-values down to, but not below, 0.001. Clustering was conducted with Ward's linkage method based on Bray-Curtis dissimilarities.

### Data and analysis availability

All raw sequence data are available on the Figshare open-access data repository under DOI: 10.6084/m9.figshare.1506779. All data analysis details and R scripts are available on GitHub: https://github.com/jfmeadow/Meadow_HumanMicrobialCloud_Analysis.

## RESULTS

Sequencing of bacterial 16S rRNA genes across the two experiments resulted in more than $14 \times 10^6$ quality-filtered sequences. Since the objective of the first experiment was to determine the detectability of a single occupant in a cleaned room, we first focused on differentiating occupied air from unoccupied.

### Human occupants shed a detectable bacterial cloud

In the first experiment, bacterial assemblages in samples from occupied and unoccupied air were significantly different, regardless of occupant, collection method or trial duration ($p = 0.001$; from PERMANOVA tests on Canberra distances; Table 1). When considering

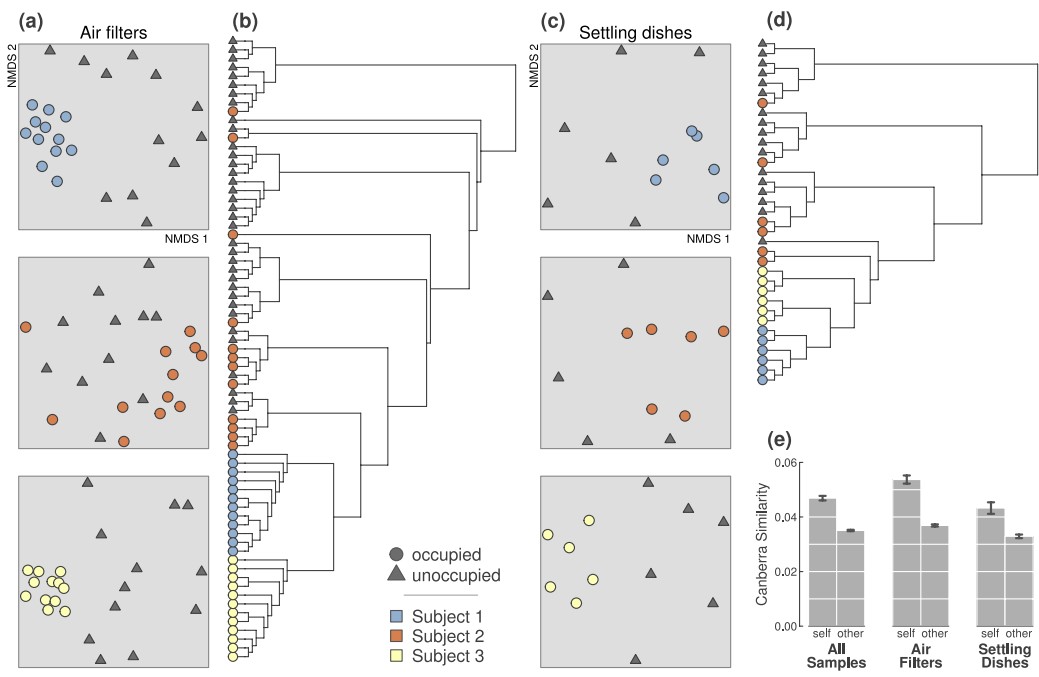

**Figure 1 Occupied and unoccupied bioaerosols during the first experiment were significantly different, and occupants were distinguishable during all 4-hour sampling periods.** (A) All three occupants were discernible from simultaneously unoccupied air (all $p$-values = 0.001). Ordination plots are 2-dimensional NMDS from Canberra distances. (B) Occupants were distinguishable from one another based on bacteria collected in air filters ($p$-value = 0.001). (C) Occupants were discernible from unoccupied samples based on bacteria collected in settling dishes ($p$-values = 0.003, 0.044, and 0.005; for Subjects 1, 2, and 3). (D) Settled particles from each occupant were somewhat less consistently identifiable, even though the three occupants were significantly different ($p$-value = 0.001). (E) Occupant microbial clouds were more similar to other samples from the same person than to other occupants, regardless of sampling method. This difference was significantly more pronounced than that of unoccupied samples taken simultaneously during sampling periods (Fig. S1). Error bars represent ±1 standard error on pairwise Canberra similarities.

individual sampling periods (Table 2 and Fig. 1), all three individuals could be clearly detected above background airborne communities after 4- and 2-hours from the airborne bacteria collected on air filters. Only Subjects 1 and 3 were consistently detectable from particles in settling dishes at both time intervals; Subject 2 was significantly detectable during the 4-hour sampling period, but not during the 2-hour sampling period ($p = 0.34$).

These community differences were evident in a few specific human-associated bacterial taxa. Indicator analysis (*Dufrêne & Legendre, 1997*) identifies those operational taxonomic units (OTUs) that significantly and consistently distinguish a given treatment, in this case bacterial groups that were especially abundant in occupied chambers or those that helped to differentiate among individual occupants. All significant indicator taxa (defined as having an indicator value >0.5 and $p$-value <0.01) detected in 4-hour air filters from the occupied chamber were closely related to human-associated bacterial taxa found in the NCBI bacterial isolate database. Conversely, all top indicator taxa from unoccupied samples were related to bacteria from non-human environments, ostensibly introduced by supply air to the climate chamber. Taxa with indicator values >0.7 are shown in Table 3.

**Table 2 During the first experiment, occupants were always detectable in air filters, and generally in settling dishes.**

| Subject | Hours | Sample type | $n$ | $R^2$ | $p$[a] |
|---|---|---|---|---|---|
| 1 | 4 | air filter | 24 | 0.061 | 0.001 |
| 2 | 4 | air filter | 24 | 0.064 | 0.001 |
| 3 | 4 | air filter | 23 | 0.051 | 0.001 |
| 1 | 2 | air filter | 24 | 0.061 | 0.001 |
| 2 | 2 | air filter | 22 | 0.05 | 0.013 |
| 3 | 2 | air filter | 23 | 0.049 | 0.006 |
| 1 | 4 | settling dish | 12 | 0.102 | 0.003 |
| 2 | 4 | settling dish | 12 | 0.098 | 0.005 |
| 3 | 4 | settling dish | 11 | 0.105 | 0.044 |
| 1 | 2 | settling dish | 12 | 0.099 | 0.005 |
| 2 | 2 | settling dish | 12 | 0.092 | 0.344 |
| 3 | 2 | settling dish | 12 | 0.097 | 0.012 |

**Notes.**

[a] Results are from PERMANOVA on Canberra distances.

**Table 3 Indicator OTUs are reflective of treatment and individual occupants from the first experiment.**

| Closest 16S NCBI isolate and accession | Indicator type[b] (and gender) | Isolate source environment | Sequence similarity (%) to NCBI isolate | Indicator value | $p$ |
|---|---|---|---|---|---|
| *Dolosigranulum pigrum* NR_026098.1 | Subject 1 (m) | Human clinical | 99 | 0.95 | 0.001 |
| *Lactobacillus crispatus* NR_074986.1[a] | Subject 3 (f) | Human vagina, gut | 100 | 0.91 | 0.001 |
| *Corynebacterium tuberculostearicum* NR_028975.1[a] | Occupied | Human sinus, skin | 100 | 0.82 | 0.001 |
| *Corynebacterium amycolatum* NR_026215.1[a] | Occupied | Human mucous, skin | 100 | 0.8 | 0.001 |
| *Corynebacterium pseudodiphtheriticum* NR_042137.1 | Subject 1 (m) | Human clinical | 100 | 0.78 | 0.001 |
| *Dietzia maris* NR_037025.1 | Subject 1 (m) | Human clinical | 95 | 0.78 | 0.001 |
| *Anaerococcus prevotii* NR_074575.1 | Subject 3 (f) | Human lung, vagina | 100 | 0.78 | 0.001 |
| *Corynebacterium mucifaciens* NR_026396.1[a] | Occupied | Human wound | 100 | 0.77 | 0.001 |
| *Staphylococcus epidermidis* NR_074995.1[a] | Occupied | Human skin | 100 | 0.77 | 0.001 |
| *Facklamia ignava* NR_026447.1 | Subject 3 (f) | Human clinical | 100 | 0.75 | 0.001 |
| *Stenotrophomonas maltophilia* NR_074875.1[a] | Unoccupied | Soil, aquatic | 100 | 0.73 | 0.001 |
| *Streptococcus oralis* NR_042927.1 | Occupied | Human oral | 100 | 0.69 | 0.002 |
| *Corynebacterium massiliense* NR_044182.1 | Subject 1 (m) | Human clinical | 100 | 0.67 | 0.001 |
| *Corynebacterium jeikeium* NR_074706.1[a] | Subject 1 (m) | Human skin | 100 | 0.66 | 0.001 |
| *Peptoniphilus ivorii* NR_026359.1 | Subject 3 (f) | Human clinical | 98 | 0.65 | 0.001 |
| *Corynebacterium simulans* NR_025309.1 | Subject 1 (m) | Human clinical | 99 | 0.64 | 0.001 |
| *Corynebacterium riegelii* NR_026434.1 | Subject 3 (f) | Human vagina | 99 | 0.63 | 0.001 |
| *Peptoniphilus harei* NR_026358.1 | Occupied | Human clinical | 100 | 0.61 | 0.001 |
| *Leuconostoc gelidum* NR_102984.1 | Subject 2 (m) | Fermented food | 100 | 0.6 | 0.001 |
| *Citrobacter freundii* NR_028894.1 | Unoccupied | Soil, aquatic | 100 | 0.6 | 0.003 |

**Notes.**

[a] Corresponding OTU was also among the most abundant and thus included in Fig. 2.

[b] Individual occupants are shown if indicator taxa were significant for that particular person, and "*occupied*" if significant for all occupants together.

## Occupants differ in their personal microbial cloud

In addition to our finding that occupants were detectable from their microbial contributions of bioaerosols and/or settled particles, bacterial assemblages were also *unique* to each of the three occupants, meaning that samples from each individual were statistically distinct and identifiable to that occupant ($p = 0.001$; from PERMANOVA on 4-hour air filters from each occupant; Fig. 1 and Table 1). Each occupant, however, was identifiable in different ways. For instance, microbial assemblages in air filters from Subject 2's 4-hour sampling periods were statistically more variable than the other two occupants ($p < 0.0001$; from ANOVA test on beta-dispersion distances); the 2-hour sampling period followed the same general pattern.

Each of the three occupants was also identifiable by distinct bacterial OTUs. For example, an OTU 99% similar to *Dolosigranulum pigrum* (Fig. S2) was similarly enriched in all of Subject 1's samples, yet it was absent for other occupants. Subject 2's samples were dominated by a *Staphylococcus* OTU (100% similar to *Staphylococcus epidermidis*; Fig. S2). Although other closely related OTUs were present throughout the study, this same OTU was less abundant in unoccupied samples, and when the chamber was occupied by anyone else. The sole female in the first experiment, Subject 3, was strongly associated with a *Lactobacillus* OTU (Fig. S2) 100% similar to *Lactobacillus crispatus*, a bacterium that is commonly found dominating healthy vaginal samples. This OTU was essentially absent throughout the rest of the first experiment. All OTUs discussed above were also significant indicator taxa (Table 3).

## Occupant microbial clouds can be detected on surrounding surfaces

In addition to airborne particles, we collected settled particles in sterile settling dishes as an estimate of the pool of potentially persistent particles. Consistent with results from air filter samples, occupied vs. unoccupied samples were always significantly different during 4-hour time periods (Tables 1 and 2), and the most abundant OTUs in each trial reflect those found in air filters (Fig. S2).

## Targeted subset of human-associated OTUs

Based on the predominance of these human-associated OTUs over background air in the first experiment, and on previous human microbiome research, we designed the second experiment and subsequent analyses around this subset of bacterial families (*Corynebacteriaceae, Staphylococcaceae, Streptococcaceae, Lactobacillaceae, Propionibacteriaceae, Peptostreptococcaceae, Bifidobacteriaceae, Micrococcaceae, Carnobacteriaceae, Dietziaceae, Aerococcaceae*, and *Tissierellaceae*). These are hereafter referred to as "targeted OTUs." Specifically, we selected these families based on three criteria: (1) OTUs representing these families were consistently significant predictors of human occupants vs background air in the first experiment (Table 3); (2) the relative abundances of these families were always elevated in occupied samples compared to unoccupied samples; and (3) all of these families are consistently detected as members of the healthy human microbiome (*HMP Consortium, 2012*; *Grice & Segre, 2011*; *Ravel et al., 2011*), and as indicators for human

occupancy in the built environment (*Fierer et al., 2010*; *Flores et al., 2011*; *Meadow et al., 2013*; *Kembel et al., 2014*; *Meadow et al., 2014*).

## Occupant identifiability

Results from the first experiment illustrated that (a) occupants emit a detectable airborne signal; (b) that signal is the result of elevated abundance of a specific set of human-associated bacterial taxa; and (c) each occupant's personalized airborne signal can be statistically differentiated from other occupants. Thus, our second experiment was focused on directly analyzing the subset of targeted human-associated airborne bacteria to determine the detectability and personalized nature of a given individual's microbial cloud. To do this, we sampled the air surrounding each of 8 occupants, as well as the supply and exhaust air moving into and out of the occupied chamber, respectively (Fig. S1B). We then analyzed the targeted subset of human-associated bacterial OTUs, described above, to determine if and how many occupants could be statistically differentiated just by the air around them.

We found that each of the eight occupants emitted their own characteristic concentration of airborne particles. These particle concentrations were correlated with the proportion of human-associated bacteria in the surrounding air, and subsequently with our ability to identify each unique occupant from their microbial cloud (Figs. 2A–2C). As before, some occupants' microbial clouds were more detectable than others, and for each person this was predicted by the proportion of targeted human-associated OTUs in an occupant's respective dataset. Samples where the targeted subset of OTUs composed more than 20% of the total generally clustered correctly by occupant, while those with less were generally unable to be classified as being from a specific occupant (Figs. 2A, 2D and 2E). The same apparent 20% threshold also applied to the human cloud signal detected in the exhaust air leaving the chamber. We were only able to classify exhaust air as coming from a particular occupant if sufficient human-associated taxa were detected, and this was only possible for two of the eight occupants. For four others, the air in the occupied chamber was distinct from background air and generally from other occupants, but they could not be detected in exhaust air, while two occupants could not be detected from airborne sources at all. Figure 3 displays three such examples of detectability in occupied air and exhaust air.

As in the first experiment, each distinguishable occupant was strongly associated with individual OTUs from human-associated taxonomic groups. For example, both of the female occupants shown in Fig. 4 (orange and red bars) were associated with OTUs related to common vaginal bacteria (*Lactobacillus crispatus* and *Gardnerella vaginalus*), mirroring the gender-relevant findings from the first experiment. Additionally, while some *Corynebacterium* and *Propionibacterium* OTUs were abundant and common among all occupants, some individual OTUs within these genera were indicative of individual occupants (Fig. 4), indicating that species- or strain-level variation in airborne bacteria can inform future microbial cloud and identifiability studies.

## DISCUSSION

Our data make clear that an occupied space is microbially distinct from an unoccupied one, and reveal for the first time that individuals occupying a space can emit their own

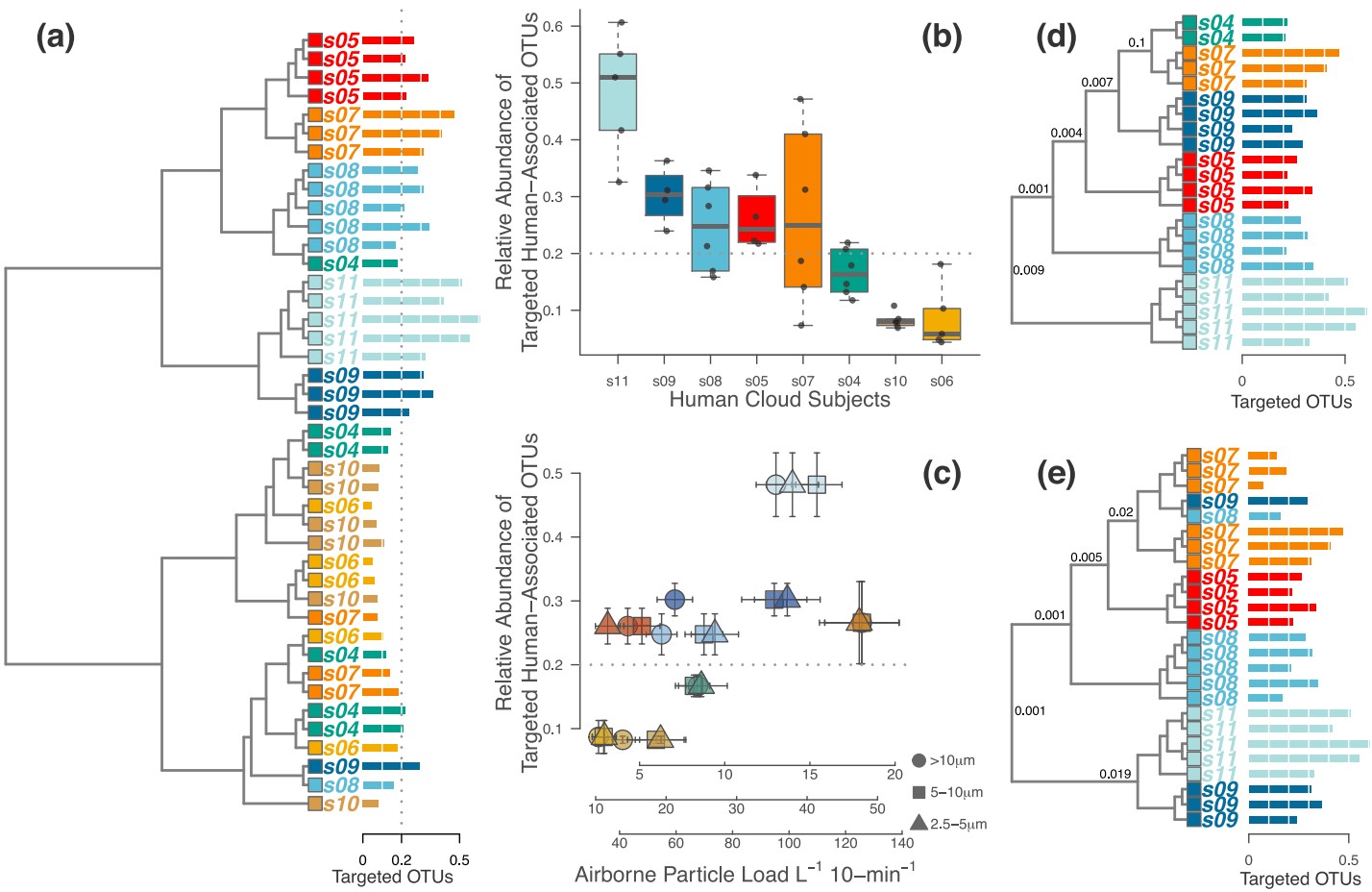

**Figure 2  Half of the occupants in the second experiment were clearly distinguishable, but this depended on the magnitude of human-associated bacteria shed during occupation.** (A) When analyzing only the targeted human-associated bacterial taxa, only samples in the top half of the dendrogram tended to be correctly classified together. Those samples that failed to cluster together were generally below the apparent 20% human-associated threshold (gray dotted line). Each tip on the tree is a separate sample from a single occupant (*subject s04-s11*). Each occupant is a different color, and the colors correspond with Figs. 3 and 4. Horizontal bars (identical to those used in D & E) show the proportion of targeted human-associated bacterial OTUs in each sample. These same values are shown as the *y*-axes in B & C. (B) Each occupant yielded a consistent proportion of human-associated taxa. (C) Airborne particle counts (*x*-axis) correlate with the proportion of airborne human-associated taxa detected around each occupant (*y*-axis). (D) When the dataset was limited to only those samples above the 20% threshold, all samples cluster appropriately by individual human subject. (E) Alternatively, if limiting the dataset to only those *occupants* whose median sample proportion was above 20%, results were nearly identical except for two misclassifications. *P*-values shown at major nodes in D & E are from PERMANOVA tests on separation of individual clades.

distinct personal microbial cloud. It is unsurprising that humans leave their microbial signature behind in the built environment (*Hospodsky et al., 2012*; *Täubel et al., 2009*; *Fierer et al., 2010*; *Flores et al., 2011*; *Flores et al., 2013*; *Meadow et al., 2013*; *Kembel et al., 2012*) or that inactive humans emit particles (*You et al., 2013*), but our study suggests that bacterial emissions from a relatively inactive person, sitting at a desk for instance, have a strong influence on the bacteria circulating in an enclosed space and on surrounding surfaces. Previous research has found that human activity in an indoor space results in the detection of human-associated airborne bacteria (*Hospodsky et al., 2012*; *Meadow et al., 2013*; *Qian et al., 2012*); this human-microbial signal is due to a combination of resuspended dust,

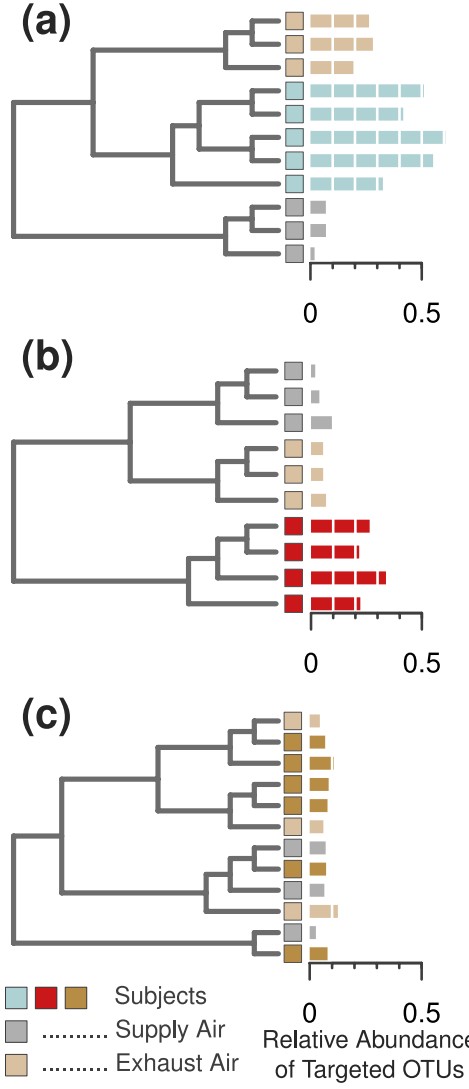

**Figure 3 Three example cases of detectability in the occupied chamber and the exhaust ventilation system.** (A) Subject 11 was an example of ideal detection in the ventilation system—we were able to find sufficient human-associated OTU concentrations to correctly classify the air leaving the occupied chamber. (B) Most occupants, however, did not emit sufficient bacterial concentrations to be detected in the ventilation system, even when they were readily detected within the occupied chamber. (C) Two subjects emitted nearly undetectable concentrations of particles (Fig. 2C) and human-associated bacterial OTUs (Fig. 2B), and were thus impossible to detect or identify in either the occupied chamber or the exhaust ventilation system.

emission from clothing, and active particle emission from occupants. In our study, we made all attempts to eliminate the potential for resuspended dust by heavily cleaning the interior of a controlled climate chamber and eliminating most movement within the chamber. We controlled for clothing-related particle emission by having all occupants wear identical, clean, newly purchased, minimal clothing (tank-top and shorts). The result is that we now have a clearer picture of individual shedding rates, personal identifiability, and the residual fate of the personal microbial cloud in the built environment.

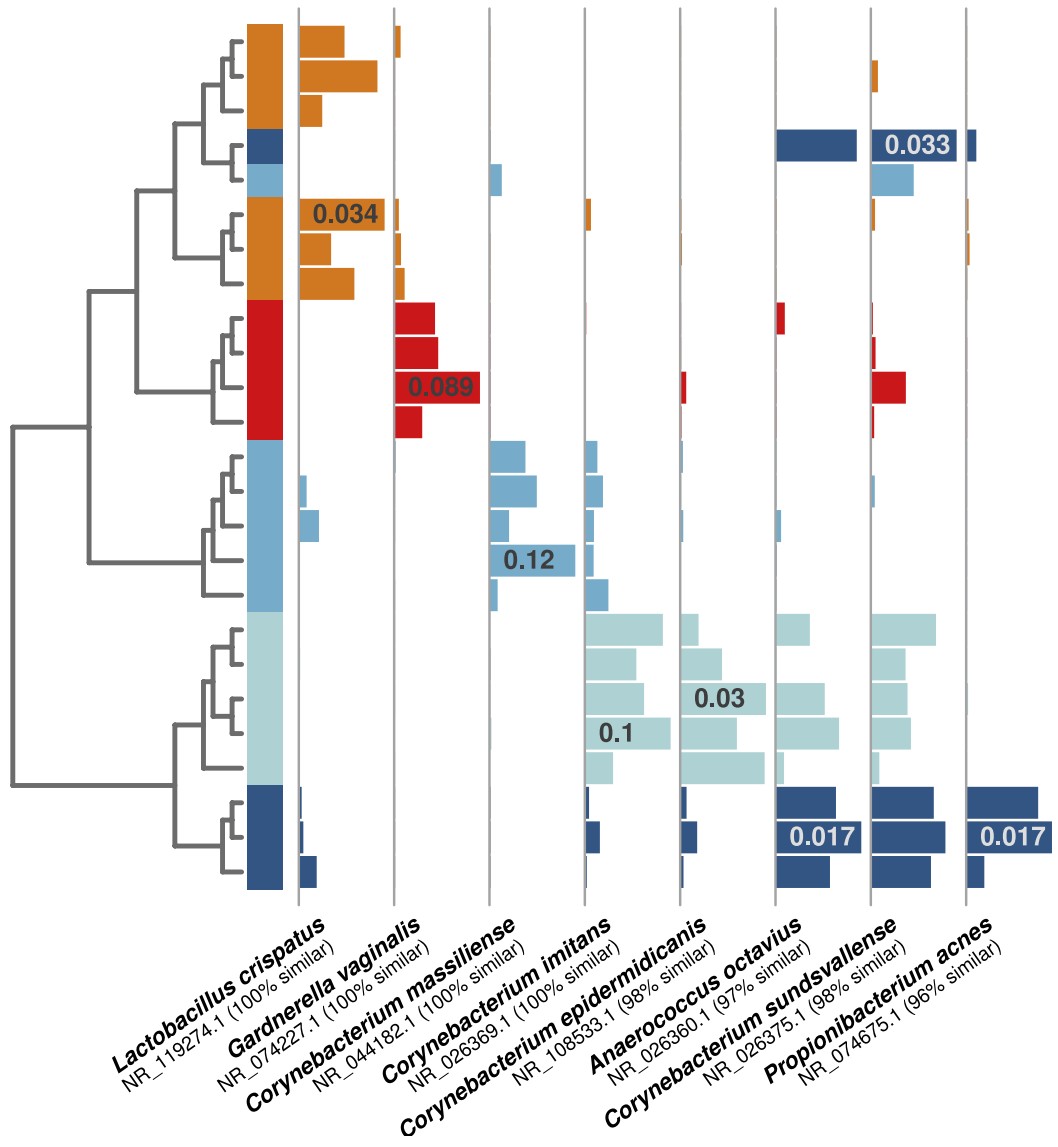

**Figure 4 Individual human-associated bacterial OTUs helped to distinguish occupants.** When considering the five most statistically distinguishable occupants (cluster diagram from Fig. 2E), each was associated with a set of significant indicator OTUs, and eight examples are shown here. Each was (A) significantly associated with an occupant (all *p*-values <0.01), (B) among the 10 most abundant for that given occupant, and (C) among the 50 most abundant targeted OTUs in the whole dataset. Horizontal bars show each OTU's relative abundance, with maximum relative abundance shown in a single bar. OTU names matched from the NCBI 16S isolate database are given below each set of bars.

Our approach was to measure several different aspects of the personal microbial cloud: airborne particle load, airborne bacterial communities, and settled bacterial communities. When we just tried to detect an occupant, all 4-hour sampling periods during the first experiment resulted in highly significant airborne community differences between simultaneously sampled occupied and unoccupied chambers (Tables 1 and 2; Fig. 1). This difference was evident regardless of which person was occupying the space, whether

air or settled particles were used, and regardless of whether occupants were analyzed alone or in combination with other occupant sampling periods. The prominent bacterial OTUs detected in each trial were clearly indicative of their ostensibly human source (Fig. S2), and this was evident in the most abundant OTUs as well as indicator taxa (i.e., those most strongly associated with each trial across replicates). When sampling time, and thus air-borne microbial biomass, was cut in half to 2-hour trials, results were generally consistent (Table 2), although with more variability among occupants, and this suggests a potential detection threshold given the current technology employed here. Approximately 1.2 m$^3$ of air passed through each sampler during the 2-hour trials, and 2.4 m$^3$ in the 4-hour trials; one m$^3$ of air can contain as many as 10$^6$ bacterial cells, but this concentration can fluctuate based on location, bioaerosol source, and ambient conditions (*Rook, 2013*; *Burrows et al., 2009*; *Tong & Lighthart, 2000*; *Lighthart & Shaffer, 1995*). It is possible that a person occupying a room for a shorter amount of time will not shed a sufficient number of bacterial particles to overcome background airborne bacterial concentrations, and thus go undetected. Although further investigation is required to fully understand the limits of human bioaerosol detection, results from our second experiment supported the assumption that detectability is a function of the amount of bacterial biomass shed by an occupant. The airborne particle load was generally predictive of each occupant's airborne load of human associated bacterial OTUs, and subsequently of each person's classifiability.

Airborne bacteria are only a short-term pool of microbes emitted by occupants in a given space; those that settle out over time have the potential to be dispersed through surface contact, or be resuspended by subsequent occupant movement. To better understand the fate of the personal microbial cloud, we also collected settled particles in sterile collecting dishes around each occupant as a proxy for the potential residual signal that an occupant might leave behind. Bacteria detected in these dishes mirrored those found in air filters (Figs. S2 and 1; Tables 1 and 2), and the same hallmarks of an occupied space were reflected in the most abundant OTUs and the most significant indicator taxa (Fig. S2 and Table 3). Results from the two different durations (2- and 4-hour sampling periods in the first experiment) were also consistent with air filter data; all three occupants in the first experiment were clearly discernible at 4-hours, while occupant signals were less pronounced after only 2 h (Table 2).

One of the most surprising results from the first experiment was the extent to which the three different occupants were easily discernible from one another, both from a microbial community perspective and also when considering individual bacterial OTUs (Figs. 1 and S2; Tables 1–3). We designed the second experiment to better understand what leads to airborne detectability and identifiability. Since the most indicative OTUs were from human-associated bacterial groups, we focused analytical efforts on just this subset of airborne bacteria. Individual occupants varied in their proportion of these targeted OTUs, from 4 to 61% in a given sample, but each occupant was generally consistent in the concentration of their own detectable microbial cloud (Fig. 2C).

Airborne particles, regardless of their biological nature, were also optically measured throughout the experiment in addition to microbial communities, to better understand

personalized particle emissions from different people (*Qian, Peccia & Ferro, 2014*). Particle emissions from the eleven occupants in this study varied substantially but were consistent for each person (Table S1 and Fig. 2C). We might expect that the occupant emitting the most particles would also be the most easily discernible from their microbial cloud. This was generally the case, since airborne particle concentrations tended to correlate with the proportion of human-associated taxa, and with personal identifiability. We did see, however, clear exceptions to this assumption. For example, Subject 3 was always consistently discernible from microbial data, and yet was nearly undetectable via airborne particle counts, while Subject 2 revealed the opposite pattern. Although this raises questions that cannot conclusively be answered here, this discrepancy might suggest that particle counts alone cannot be used as a proxy for personal microbial cloud identifiability, but rather reflect interpersonal variation in hygiene, skin health, respiration and perspiration rates, or other occupant characteristics that should be investigated in the future.

The potential identifiability we report for individual personal microbial clouds clearly suggests a forensic application for indoor bioaerosols, for example to detect the past presence of a person in an indoor space. Such applications will certainly require further research; the patterns we found are likely to be more nuanced in a crowd of occupants, in a larger indoor space, or in the presence of resuspended dust. Personal classifiability in our study was likely dependent on relatively low background microbial biomass (e.g., dust) in our experimental chamber, and these patterns were not evaluated in the presence of multiple occupants, similar to the recent surface identifiability study from Fierer and colleagues (*2010*). However, unlike identifiability after surface contact, the personal microbial cloud is highly ephemeral, such that detection of an occupant after they have left a space will almost inevitably rely on either settled particles or capture in ventilation systems. To this point, we found that settled particles revealed occupant individuality, and that at least two people were detectable in exhaust air leaving the occupied chamber. Notably, when air exchange rates were increased from 1 to 3 ACH, detectibility and identifiability are much more difficult due to increased dilution with background air (Fig. S4). This suggests important applications for understanding the impact of ventilation on person-to-person microbial transmissions in health-care facilities, or during disease outbreaks in the built environment. Ventilation has long been acknowledged as important for indoor disease transmission, and our findings suggest that increasing air flow rates from 1 to 3 ACH nearly eliminates the detectible human microbial cloud.

The eleven different occupant trials were each conducted on different days, requiring occupation of the chamber by a different person each day. Short-term temporal bioaerosol dynamics have been detected in previous studies (*Meadow et al., 2013*; *Bowers et al., 2013*), and we did find marginal differences among the background bioaerosols during both experiments reported here. However, the day-to-day differences among unoccupied sampling periods in the first experiment, and among supply air samples from different days in the second, was much smaller in all instances than the differences among occupied samples (Fig. S3). Nor was day-to-day variation accompanied by significant indicator taxa differentiating unoccupied samples from one day to the next, whereas most occupants

were personalized in their emitted indicator taxa. Furthermore, the most abundant OTUs detected in occupied samples changed along with the occupants, while the same OTUs were most abundant in unoccupied samples each day regardless of when they were taken. Thus day-to-day temporal dynamics were less substantial than the clear difference we observed among occupants.

As humans we spend a substantial portion of our lives indoors, up to 90% in industrialized nations (*Klepeis et al., 2001*), and human density in urban areas is expected to increase. While indoors, we are constantly interacting with microbes other people have left behind on the chairs in which we sit, in dust we perturb, and on every surface we touch. These human-microbial interactions are in addition to the microbes our pets leave in our houses, those that blow off of tree leaves and soils, those in the food we eat and the water we drink. It is becoming increasingly clear that we have evolved with these complex microbial interactions, and that we may depend on them for our well-being (*Rook, 2013*). It is now apparent, given the results presented here, that the microbes we encounter include those actively emitted by other humans, including our families, coworkers, and perfect strangers.

## ACKNOWLEDGEMENTS

We thank members of the Biology and the Built Environment Center for their input in experimental design, and Energy Studies in Buildings Laboratory for assistance at their climate chamber facility. We also thank Dr. Seema Bhangar for her advice in collecting and analyzing particle data. Finally, we thank 11 anonymous volunteer human occupants for shedding their bioaerosols into our sampling apparatuses.

### Funding

This work was funded by a grant to the Biology and the Built Environment Center from the Alfred P. Sloan Foundation Microbiology for the Built Environment Program. The funders had no role in study design, data collection and analysis, decision to publish, or preparation of the manuscript.

### Grant Disclosures

The following grant information was disclosed by the authors:
Alfred P. Sloan Foundation Microbiology.

### Competing Interests

Jessica Green is an Academic Editor for PeerJ.

### Author Contributions

- James F. Meadow conceived and designed the experiments, performed the experiments, analyzed the data, wrote the paper, prepared figures and/or tables, reviewed drafts of the paper.
- Adam E. Altrichter and Ashley C. Bateman conceived and designed the experiments, performed the experiments, reviewed drafts of the paper.

- Jason Stenson conceived and designed the experiments, performed the experiments, analyzed the data, prepared figures and/or tables, reviewed drafts of the paper.
- GZ Brown conceived and designed the experiments, contributed reagents/materials/analysis tools, reviewed drafts of the paper.
- Jessica L. Green and Brendan J.M. Bohannan conceived and designed the experiments, performed the experiments, contributed reagents/materials/analysis tools, reviewed drafts of the paper.

### Human Ethics

The following information was supplied relating to ethical approvals (i.e., approving body and any reference numbers):

The subjects were informed as to the full nature and design of the study and gave written consent to be participants. All research protocols were approved by the University of Oregon Institutional Review Board (protocol # 03172014.021). Identities of participants were never recorded on samples or in resulting datasets.

### DNA Deposition

The following information was supplied regarding the deposition of DNA sequences:

All sequence data used in this paper have been deposited in the open access data repository Figshare:

10.6084/m9.figshare.1506779.

All scripts used in statistical analysis can be found in GitHub:

https://github.com/jfmeadow/Meadow_HumanMicrobialCloud_Analysis.

### Supplemental Information

Supplemental information for this article can be found online at http://dx.doi.org/10.7717/peerj.1258#supplemental-information.

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
