# Peer review of "Humans differ in their personal microbial cloud"

_PeerJ, doi:10.7717/peerj.1258_

## Round 0.1 · original submission · Minor Revisions

This is a very good work, all reviewers agree that it should be accepted, one of them loves it, another accepted with minor revisions, the last one is not pleased with the writing style and presentation (that should be addressed). However, since none of them, or the editor, has any problem with the science, I think the authors should follow all the possible recommendations in order to have a better manuscript. I look forward to seeing your revised version.

Reviewer 1 ·

Basic reporting

Please make public the data sets and include the accession numbers.

Experimental design

No Comments

Validity of the findings

No Comments

Additional comments

This paper shows an excellent experimental design and analysis. It has been a pleasure to have the opportunity to read it. I have a few minor comments.

It would be important to reproduce the results to be more explicit about how the statistical analysis was done, particularly PERMANOVA and NMDS. Both analyzes are complex and essential to understand the work. Please consider sharing your scripts.

Table 1 and 3. Clearly indicate the difference between "occupied" and "3 people".

Legend of figure 2C. It says “predict” which is inconsistent with what is mentioned in the discussion (lines 210-222). Change it for a less categorical phrase.

Reviewer 2 ·

Basic reporting

The paper is clearly written and includes sufficient information in the introduction and in all other sections.
I found the text somehow arcane, being myself not an expert in bioinformatics. Nonetheless I believe the text is generally consistent and mostly not repetitive.

Experimental design

The work described is clearly original, relevant and meaningful, representing the novel ways where current DNA sequencing technology can be used. In this case to investigate the microbial environment in closed spaces, as related to human occupancy.

Validity of the findings

I can not offer a strong review on the soundness of the detailed protocols. In general terms they appear controlled and properly executed.
The conclusions are appropriate and justified in my view.

Additional comments

The manuscript reports a very interesting enquire on the nature of the microbial cloud emmited by humans, including its differentiating capacity. It consists on initial observations that appear sound and should serve as the basis for future research.

Reviewer 3 ·

Basic reporting

This is a fascinating study done to characterize the sometimes person-specific microorganisms shed from the human body. Unfortunately, the manuscript was not well written and consequently it was difficult to understand.

As the authors explain the influence of humans on the built environment has been studied by others and a fair amount is known. So I found the goal stated in the Introduction — to understand the human contribution to bioaerosols within built environments (lines 46-49) — to be vague and wanting. Similarly, after reading the first paragraph of the Discussion I was unsure which findings in the present study were truly novel. The authors should try harder to articulate the specific questions addressed and to describe the knowledge gap filled by the present study.

Experimental design

The means used to collect samples (sanitized experimental climate chambers) allowed sampling of intake and exhaust air as well as settled particles and seem perfectly suited for the task. The experiments done were reasonable and the data were suitably analyzed.

It is a bit curious that the skin of subjects themselves was not sampled. It might answer the question of whether the organisms found in shed material reflect those found on the skin itself. It is presumed that many of the organisms in microbial cloud are derived from the skin, but this simple analysis would have provided insight to the source of organisms.

Validity of the findings

No concerns.

Additional comments

The manuscript is not well written and the Tables and Figures are not carefully prepared. Here are some examples:

⁃ There are various problems with syntax that should be addressed. Some examples are pointed out in MINOR COMMENTS below. The manuscript should be carefully edited to increase the precision of these problem sentences.

⁃ Figure 1. The figure legend should better explain the figure. For example, are ordination plots shown in this figure?

⁃ Table 1 and 2 desperately need greater transparency and footnotes to explain abbreviations, “n”, “R^2”, and so forth.

⁃ Table 3. Heading of column 1 is jargon and incomplete. Were these actually isolates? Is it correct to refer to isolated and named bacterial species as OTUs? (I don’t think so.) Similarly, does it make sense to refer to a “Staphylococcus OTU”? (I think an OTU is a gene sequence is that has similarity to a known and named genus or species.) Also see line 113 and other instances in the text.

- Figure 2. The information in this figure is dense and not readily understood. The authors could help by explaining the graphics better in the figure legend. For example, what do the different colors represent? What are the samples being clustered in panel A — for example what is “s05” and why does it show up four times? What do the circles, triangles and squares in panel C represent?

⁃ Simplify the text — some sentences are far too complex. For example, see lines 121-125, 130-133, 170-173, and 195-198.

OTHER MINOR COMMENTS:

a) Lines 19-21. The sentence is a bit clumsy. Also, what is “active human emission”?

b) Lines 21-22. References misplaced in sentence. Also, somewhere in the manuscript the authors should explain how the present work builds upon and/or differs from these previous studies? Is the present work novel and, if so, how?

c) Line 38. “Detectable human bacterial signal”. Syntax problem.

d) Line 58-59. “Interpersonal nature”? A bit mysterious.

e) Line 68. “occupied and unoccupied samples”? Syntax problem.

f) Line 70-71. “when considering individual sampling periods” and “all three individuals could be detected”. Syntax problems.

g) Figure 1. “simultaneous unoccupied air”. Syntax problem.

h) Figure 2 legend. “Depended on the proportion of human-associated bacteria shed”. Syntax problem.

---

## Round 0.2 · accepted · Accept

The authors have carefully addressed all the concerns raised by the 3 reviewers and myself. The manuscript should be accepted in its present form.

---

## Author Rebuttal · Round 0.2

Dear Dr. Souza,

We wish to submit our revised manuscript 'Humans differ in their personal microbial cloud.' We have addressed all reviewer comments (detailed below), and we think this has improved the quality of the manuscript. Specifically, we tried to expand on the novelty of the work, and carefully edited the manuscript for clarity, as per comments from Reviewer #3.

Thank you for your consideration, and we look forward to your decision.

Sincerely,
James Meadow and coauthors
* * *
# Reviewer Comments

## Reviewer 1 (Anonymous)

### Basic reporting

Please make public the data sets and include the accession numbers.

> *We have released the data and associated scripts, but now it is more clearly specified in the text.*

### Comments for the author

This paper shows an excellent experimental design and analysis. It has been a pleasure to have the opportunity to read it. I have a few minor comments.

> *Thank you.*

It would be important to reproduce the results to be more explicit about how the statistical analysis was done, particularly PERMANOVA and NMDS. Both analyzes are complex and essential to understand the work. Please consider sharing your scripts.

> *We have released all analysis scripts as a reproducible workflow. We reported this*

*in the original submission process, but we've now added the information to the manuscript itself.*

Table 1 and 3. Clearly indicate the difference between "occupied" and "3 people".

"

*Good point. Done.*

Legend of figure 2C. It says "predict" which is inconsistent with what is mentioned in the discussion (lines 210-222). Change it for a less categorical phrase.

"

*Agreed. Done.*

## Reviewer 2 (Anonymous)

### Basic reporting

The paper is clearly written and includes sufficient information in the introduction and in all other sections. I found the text somehow arcane, being myself not an expert in bioinformatics. Nonetheless I believe the text is generally consistent and mostly not repetitive.

### Experimental design

The work described is clearly original, relevant and meaningful, representing the novel ways where current DNA sequencing technology can be used. In this case to investigate the microbial environment in closed spaces, as related to human occupancy.

### Validity of the findings

I can not offer a strong review on the soundness of the detailed protocols. In general terms they appear controlled and properly executed. The conclusions are appropriate and justified in my view.

### Comments for the author

The manuscript reports a very interesting enquire on the nature of the microbial cloud emmited by humans, including its differentiating capacity. It consists on initial observations that appear

sound and should serve as the basis for future research.

## Reviewer 3 (Anonymous)

### Basic reporting

This is a fascinating study done to characterize the sometimes person-specific microorganisms shed from the human body. Unfortunately, the manuscript was not well written and consequently it was difficult to understand.

As the authors explain the influence of humans on the built environment has been studied by others and a fair amount is known. So I found the goal stated in the Introduction — to understand the human contribution to bioaerosols within built environments (lines 46-49) — to be vague and wanting. Similarly, after reading the first paragraph of the Discussion I was unsure which findings in the present study were truly novel. The authors should try harder to articulate the specific questions addressed and to describe the knowledge gap filled by the present study.

### Experimental design

The means used to collect samples (sanitized experimental climate chambers) allowed sampling of intake and exhaust air as well as settled particles and seem perfectly suited for the task. The experiments done were reasonable and the data were suitably analyzed.

It is a bit curious that the skin of subjects themselves was not sampled. It might answer the question of whether the organisms found in shed material reflect those found on the skin itself. It is presumed that many of the organisms in microbial cloud are derived from the skin, but this simple analysis would have provided insight to the source of organisms.

"

> *Human shedding to the built environment is complex, and major contributing sources likely vary from person to person -- our analysis supports this. While skin and oral swabs might have been a useful way to link individuals and their sources, they are also somewhat more invasive for volunteers. We instead focused on eliminating every possible confounding variable to characterize the environment around the person rather than the person. We agree that more in-depth characterization of body-site sources will be important as this research moves forward.*

### Validity of the findings

No concerns.

**Comments for the author**

The manuscript is not well written and the Tables and Figures are not carefully prepared. Here are some examples:

There are various problems with syntax that should be addressed. Some examples are pointed out in MINOR COMMENTS below. The manuscript should be carefully edited to increase the precision of these problem sentences.

"

*We have now carefully edited the manuscript.*

Figure 1. The figure legend should better explain the figure. For example, are ordination plots shown in this figure?

"

*Yes they are, and they are now described in more detail.*

Table 1 and 2 desperately need greater transparency and footnotes to explain abbreviations, "n", "R^2", and so forth.

"

*The footnote on each table denotes that all tests are based on PERMANOVA from Canberra Distances. 'n' and 'R^2' are used and reported consistently with the common usage of the PERMANOVA method, a method that is quite frequently employed in multivariate community analysis. We feel the provided footnote gives readers all pertinent information for investigating methods. Additionally, we have already released all analysis scripts and data with this manuscript, in case readers desire more in-depth explanation of any statistics or their underlying data.*

Table 3. Heading of column 1 is jargon and incomplete. Were these actually isolates? Is it correct to refer to isolated and named bacterial species as OTUs? (I don't think so.) Similarly, does it make sense to refer to a "Staphylococcus OTU"? (I think an OTU is a gene sequence is that has similarity to a known and named genus or species.) Also see line 113 and other instances in the text.

> *As explained in the methods section, yes these reference names and accession numbers are isolates from the curated NCBI 16S database. The heading for column 1, we feel, is succinct and conveys the correct information. As to the second point, an OTU describes a wide array of taxonomic frameworks, and is a relative term. The methods section clearly describes the analysis and cutoffs we used to cluster and define OTUs. In this table, we are referring to our sequence based OTUs that were closely related to known isolates from the NCBI database. This is a common method that we have published several times in the past with little fanfare. However, since it was initially confusing, we've clarified the distinction in the text and footnotes.*

Figure 2. The information in this figure is dense and not readily understood. The authors could help by explaining the graphics better in the figure legend. For example, what do the different colors represent? What are the samples being clustered in panel A — for example what is "s05" and why does it show up four times? What do the circles, triangles and squares in panel C represent?

> *We have clarified the figure legend. However, regarding symbols used in panel C, the x-axis is literally the legend for those symbols. We are confident that, after another quick consideration, the reviewer will agree that panel C is sufficiently decoded.*

Simplify the text — some sentences are far too complex. For example, see lines 121-125, 130-133, 170-173, and 195-198.

> *Indeed. Each example has been revised, and the manuscript has been carefully edited for clarity.*

**OTHER MINOR COMMENTS:**

Lines 19-21. The sentence is a bit clumsy. Also, what is "active human emission"?

*Reasonable question. Revised.*

Lines 21-22. References misplaced in sentence. Also, somewhere in the manuscript the authors should explain how the present work builds upon and/or differs from these previous studies? Is the present work novel and, if so, how?

*References have been deployed more appropriately, and the novelty of this work has been spelled out more clearly in the text.*

Line 38. "Detectable human bacterial signal". Syntax problem.

*Revised.*

Line 58-59. "Interpersonal nature"? A bit mysterious.

*Revised.*

Line 68. "occupied and unoccupied samples"? Syntax problem.

*Revised.*

Line 70-71. "when considering individual sampling periods" and "all three individuals could be detected". Syntax problems.

*The sentence was revised above.*

Figure 1. "simultaneous unoccupied air". Syntax problem.

"

*Revised.*

Figure 2 legend. "Depended on the proportion of human-associated bacteria shed". Syntax problem.

"

*Revised.*